# Warning Models for Landslide and Channelized Debris Flow under Climate Change Conditions in Taiwan

Ho-Wen Chen [1,2] and Chien-Yuan Chen [3,*]

1    Department of Environmental Science and Engineering, Tunghai University, Taichung City 407705, Taiwan; hwchen@thu.edu.tw
2    Center for Smart Sustainable Circular Economy, Tunghai University, Taichung City 407705, Taiwan
3    Department of Civil and Water Resources Engineering, National Chiayi University, Chiayi City 60004, Taiwan
*    Correspondence: chienyuc@mail.ncyu.edu.tw; Tel.: +886-5-2717686

**Abstract:** Climate change has caused numerous disasters around the world. It has also influenced the climate of Taiwan, with urban areas exhibiting a temperature increase by 1 °C between 1998 and 2020. In this study, climate change and landslides in Taiwan were statistically analyzed. Cumulative annual precipitation in mountain watersheds in central Taiwan exhibit a declining trend and is lower than that in urban areas. The relatively few typhoons reduced the distribution of rainfall in mountain watersheds and fewer landslides. From 2017 to 2020, typhoon-induced rains caused fewer landslides than did other climate events such as the meiyu front, tropical low pressure, and southwesterly flow events. Three rainfall characteristics of landslide initiation were identified: high rainfall intensity over a short duration (<12 h), high-intensity and prolonged rainfall, and high cumulative rainfall over a long duration (>36 h). Combinations of warning models for landslides in cumulative rainfall–duration plots with rainfall intensity classification and mean rainfall intensity–duration plots with cumulative rainfall classification were presented. In recent (2018–2020) years, climate change has resulted in higher temperatures, less rainfall in mountain watersheds, and a lower rainfall threshold at which landslides are initiated by non-typhoon climate events.

**Keywords:** climate change; landslide; debris flow; rainfall threshold



## 1. Introduction

According to the World Meteorological Organization [1], the highest increase in the global average temperature since the preindustrial period was 1.1 °C (2015–2019), which is 0.2 °C higher than the increments in global average temperature in 2011–2015. The changing climate is rising global temperatures, shifting rainfall patterns, and more heavy rainstorms and record high temperatures [2]. Rising temperatures will increase evaporation that result in more frequent and intense storms [3]. Storm-affected areas are likely to experience increases in precipitation and increased risk of landslide. Climate change increases the risk of extreme weather events, such as droughts, flooding, and heat waves [1]. The landslide magnitudes triggered by Taiwan-invading high-level typhoons (top 5–15%) and extreme-level typhoons (top 5%) will increase by 125–200% and 77%, respectively, under climate change conditions in the Xindian River catchment in Taiwan [4]. Changes in the global climate have caused numerous disasters worldwide, and the frequency of debris flow events has increased. Debris flow hazards in the Minjiang River basin in China are expected to increase in the future due to an increase in heavy rainfall events caused by climate change [5]. The short-duration extreme rainfall events will become stronger, especially for 1-h duration events in Taiwan [6]. Climate change might lead to a rise in the overall magnitude of debris flows as a result of the larger volumes of sediments delivered to channels and an increase in extreme precipitation events by the mid-21st century [7].

Rainfall contributes to landslides, and establishing the rainfall thresholds for landslides show an unavoidable uncertainty [8]. Empirical models of rainfall thresholds for landslides

are more suitable for development on the regional scale [9]. Rainfall thresholds can be calibrated based on historical landslide data, and rainfall can be predicted or observed in real time [10]. Measurement of rainfall is relatively easy, inexpensive, and can be performed on any scale; empirical rainfall thresholds are frequently used in early warning systems for landslides and have proved valuable in predicting their occurrence [11–14].

For establishing the rainfall thresholds for landslides, five types of empirical models can be employed [14]: (i) precipitation intensity–duration (I–D) thresholds [13,15–17]; (ii) daily precipitation and effective antecedent rainfall [18,19]; (iii) cumulative precipitation–duration thresholds (Ac–D) [8]; (iv) cumulative precipitation–average rainfall intensity (Ac–Im) thresholds [20]; and (v) a combination of cumulative rainfall threshold, I–D thresholds, and antecedent soil moisture [11]. Im–D and Ac–D models are most commonly used for establishing rainfall thresholds for delivering real-time warnings. Im–D accurately reflects the characteristics of high-intensity rainfall over a short duration, whereas Ac–D accurately reflects the characteristics of high cumulative rainfall over a long duration [21]. However, in recent years, empirical lower-bound rainfall thresholds have not been able to reflect the rainfall characteristics and channelized debris flows caused by climate change.

Given that Taiwan is located in the paths of typhoons occurring in the Pacific Ocean, these storms frequently make landfall on the island. In Taiwan, debris flows have caused numerous disasters since the 1999 Chi-Chi earthquake. Moreover, debris flows induced by torrential rains caused by typhoons or southwesterly flow events are common. Rainfall characteristics that initiate debris flows include high-intensity rainfall, high cumulative rainfall, and a combination of both [22]. In one study, a three-dimensional Im–Ac–D rainfall regression surface analysis combining Ac–D and Im–Ac rainfall regression analyses was performed and the results were converted into multiple Im–D rainfall diagrams containing various cumulative rainfall quantities to determine the rainfall characteristics necessary to initiate landslides [21]. Multiple rainfall thresholds enable the identification of a set of cumulative rainfall thresholds, each corresponding to a specific landslide magnitude [23].

In the first investigation conducted in 1996, 485 creeks in Taiwan that had the potential to initiate debris flow were identified according to Taiwan's Soil and Water Conservation Bureau (SWCB; https://www.swcb.gov.tw/ (accessed date 30 April 2020)). The threshold basin area and streambed slope incline required to initiate debris flow were 0.5 km$^2$ and >15°, respectively. With the increase in the number of debris flow disasters after the Chi-Chi earthquake and typhoons Toraji and Nari in 2001, the number of debris flow–prone creeks increased substantially to 1420; accordingly, the threshold basin area required to initiate debris flow decreased to 0.3 km$^2$. Another notable change occurred between 2003 and 2012, triggered by typhoons Mindulle and Morakot in 2004 and 2009, respectively, which caused numerous debris flow disasters. Since 2016, relatively fewer typhoons have made landfall in Taiwan and the increase in the number of debris flow–prone creeks induced by southwesterly flow events has declined in magnitude (Figure 1). The objectives of the study are setting rainfall warning models to enhance the rainfall characteristics to trigger landslides and channelized debris flows under climate change by events collection, rainfall parameters measurement and statistical regression analysis.

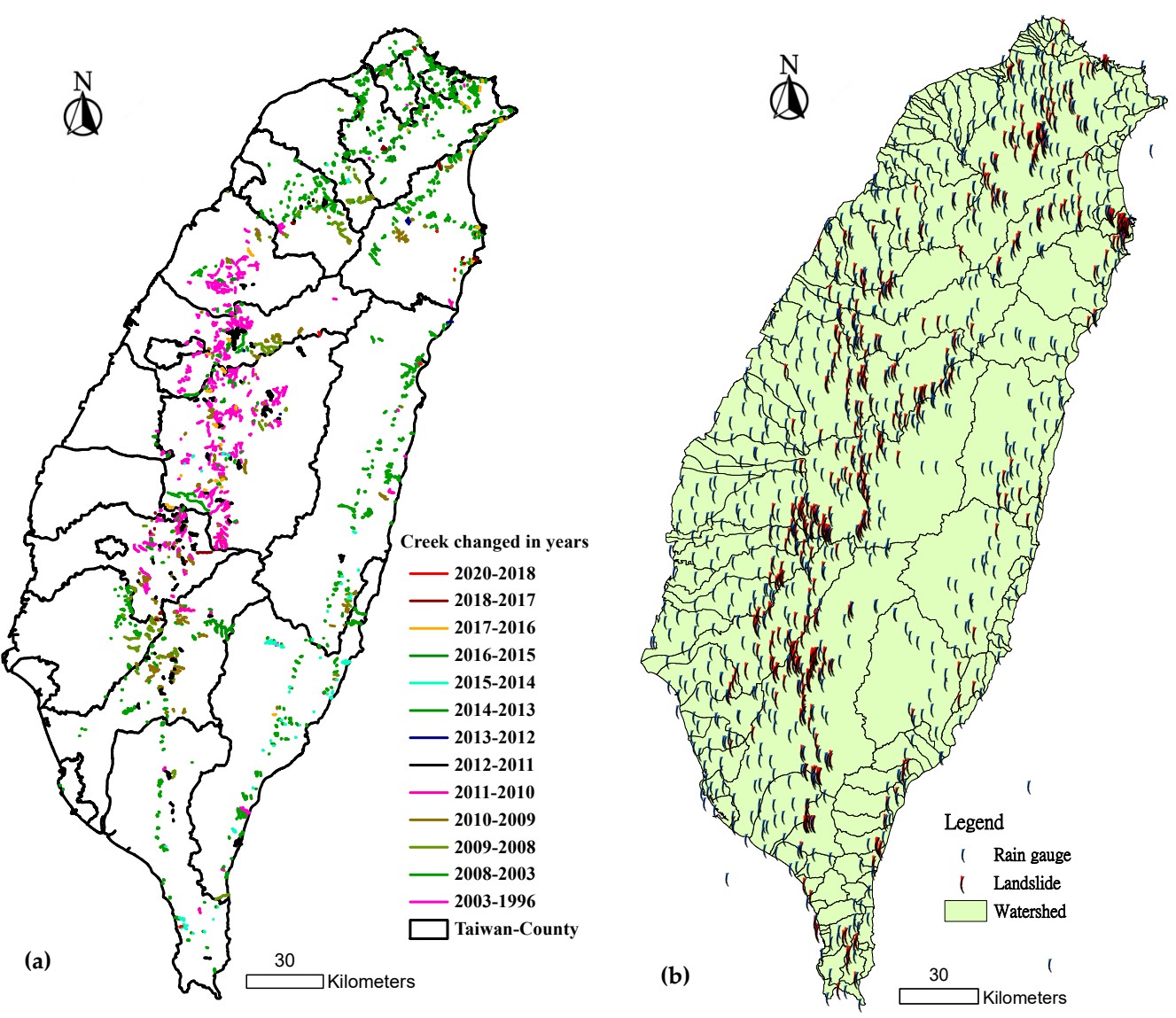

**Figure 1.** (**a**) Study area and locations of debris flow–prone creeks in Taiwan between 1996 and 2020; (**b**) locations of weather or rain gauge stations and landslides occurring between 2006 and 2020.

## 2. Materials and Methods

More than 70% of Taiwan's slopeland has an elevation of over 100 m, and 268 mountains have an elevation exceeding 3000 m. As presented in Figure 1a, debris flow–prone creeks are located on either side of the Central Mountain Ridge. Temperature and rainfall data from 25 weather stations (including 13 stations in urban areas located in plains were sourced from the Central Weather Bureau (CWB; https://www.cwb.gov.tw/ (accessed date 30 May 2020)) and those for 240 rain gauge stations located in mountain watersheds were obtained from the Water Resources Agency (WRA; https://eng.wra.gov.tw/ (accessed date 30 May 2020); Figure 1b). Data on debris flow–prone creeks and landslides between 2006 and 2020 were sourced from the SWCB. The landslides were mainly classified as shallow and deep slides, falls, and channelized debris flows. The triggering mechanisms of landslides are different from channelized debris flows (Figure 2). The available warning models are not well separated for issuing the different trigger mechanisms of landslides.

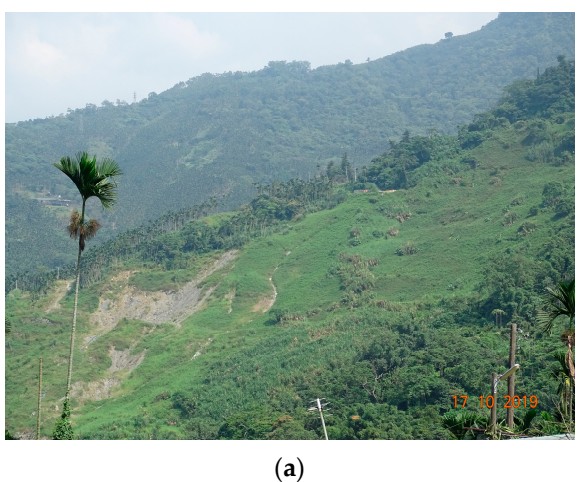
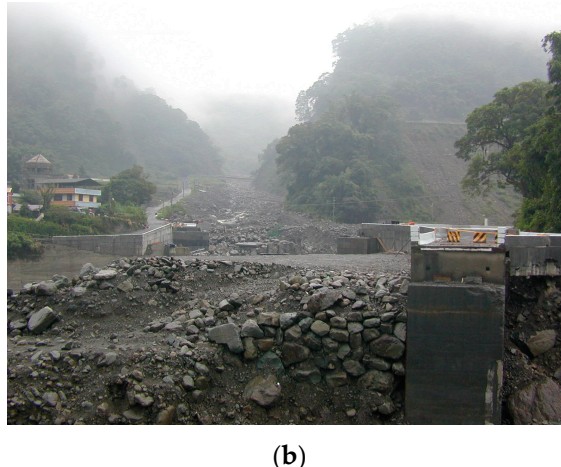

(**a**)                                                                                     (**b**)

**Figure 2.** (**a**) An example case of rainfall-induced landslide; (**b**) an example case of rainfall-induced channelized debris flow.

Spatial geographic information system analysis was performed using GIS package [24]. Data on 486 landslides, including the time of the event and the corresponding rainfall event, are presented in Figure 1 along with the rainfall duration, intensity, and effective cumulative rainfall. The effective accumulated rainfall Ac (mm) was defined as the sum of current daily rainfall $d_0$ and 2 weeks (14 days) of antecedent rainfall [25]:

$$Ac = d_0 + \sum_{i=1}^{14} \alpha_t d_t \text{ and} \alpha_t = 0.5^{t/T} \tag{1}$$

Various rainfall parameters were combined for the regression lines presentation, including mean intensity, effective cumulative rainfall, and duration. The mean rainfall intensity Im (mm/h) was defined as the average rainfall intensity until the initiation of landslide:

$$Im = Ac/D \tag{2}$$

Figure 3 displays the locations of the landslides and the corresponding rainfall conditions between 2006 and 2020 according to Im–D and Ac–D distributions. Linear and nonlinear best-fit regression analyses were used to determine the regression lines for reflecting the rainfall characteristics that trigger landslides and channelized debris flows.

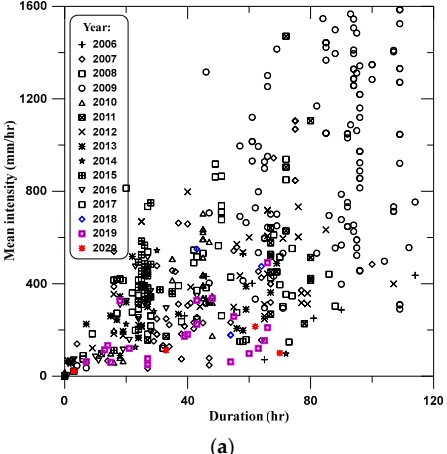
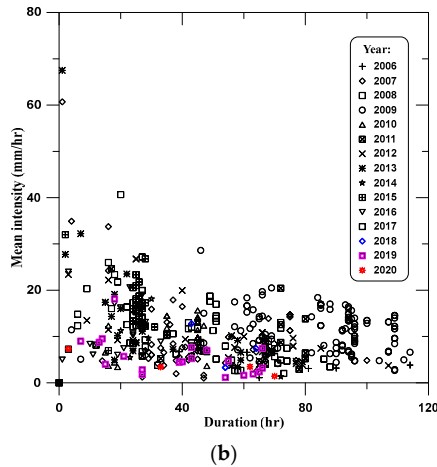

(**a**)                                                              (**b**)

**Figure 3.** Data on landslides and corresponding rainfall conditions in Taiwan between 2006 and 2020: (**a**) plot of effective cumulative rainfall versus duration; (**b**) plot of mean rainfall intensity versus duration.

Warning models for landslides in cumulative rainfall–duration plots with rainfall intensity classification and mean rainfall intensity–duration plots with cumulative rainfall classification. The potential application of the proposed rainfall warning model was evaluated using the Xiaolin landslide case and a channelized debris flow case in southern Taiwan. The objective of this study was to identify the spatial distribution of the rainfall characteristics that trigger landslides and channelized debris flows.

## 3. Results

### 3.1. Indicators of Climate Change in Taiwan

The trend of temperature change in Taiwan coincides with that of global climate change, however, the temperature increase trend in Taiwan is higher. From 1900 to 2012, the average temperature in Taiwan increased by 1.25 °C–1.5 °C in 13 urban areas in the plains [26]. Since 1998, the average temperature in Taiwan has increased considerably from the average of 23.63 °C. According to the CWB, the average temperature in urban areas was 24.56 °C in 2019 and 24.6 °C in 2020, higher by 0.34 °C and almost 1 °C compared with the corresponding temperatures in 2018 and 1998, respectively. In 2020, the average temperature was the highest among the available recorded temperatures from 1947.

Annual precipitation data indicated a periodic alteration in cumulative rainfall in recent years. As presented in Figure 4a, urban areas exhibited a lower annual cumulative rainfall than did mountain watersheds. Between 1947 and 2020, the average rainfall in 13 urban areas on the plains was 2207 mm. A high level of cumulative rainfall in plain areas has caused numerous disasters, such as Typhoon Zeb in 1998, the southwesterly current event on 12–16 June 2005; the flood event of 10 June 2012; and the flooding induced by Typhoon Megi on 27 September 2016.

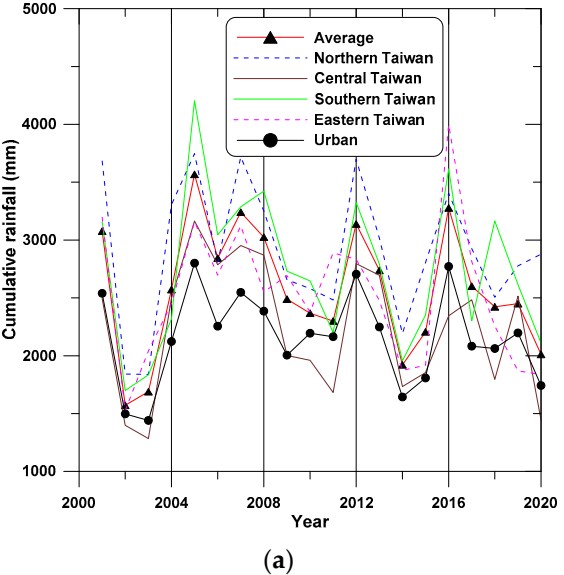
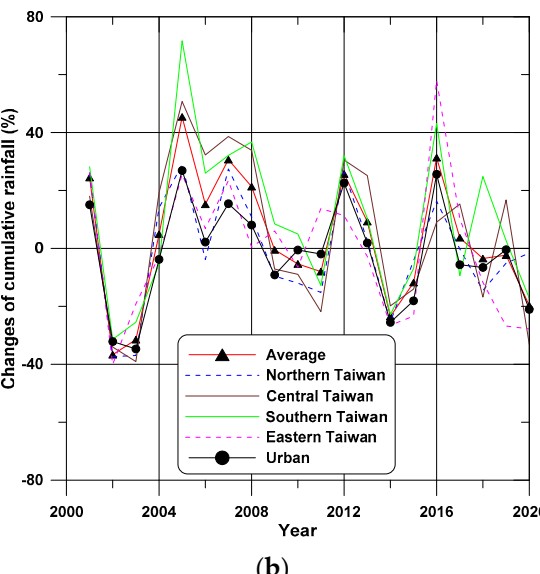

(**a**)        (**b**)

**Figure 4.** Annual cumulative precipitation between 2000 and 2020 in (**a**) 13 urban areas and mountain watersheds in northern, central, eastern, and southern Taiwan; (**b**) annual changes in cumulative precipitation, expressed as percentages.

The rainfall data for plain areas were sourced from stations located in 13 urban areas: Keelung, Taipei, Tamsui, Hsinchu, Taichung, Tainan, Kaohsiung, Hengchun, Dawu, Taidong, Chenggon, Hualien, and Yilan. In general, annual cumulative rainfall in mountain watersheds was higher than that in urban areas in the plains because of the higher elevation of the mountain watersheds. Mountain watersheds in central Taiwan received less rainfall than did urban areas, especially in 2020. As displayed in Figure 4b, mountain watersheds in northern areas received more rainfall than did other areas, and a large change was observed in southern areas.

Notably, Taiwan's topography strongly influences rainfall distribution (Figure 4). From 1897–2020, the average annual precipitation in mountain watersheds was 2507 mm [27]. In 2020, Taiwan's northern, central, southern, and eastern mountain watersheds recorded 2878, 1444 (67% of the average value), 2110, and 1837 mm of rainfall, respectively. Levels of rainfall induced by the northeast monsoon in winter and by southwesterly flows in summer were high on the windward slope. The central mountain watershed demonstrated the lowest rainfall and shortage in water supply for industries and domestic usage by dams in 2020.

Typhoons are the main source of rainfall in Taiwan, followed by the meiyu front, tropical low pressure, and southwesterly flow events [28]. Rainfall induced by typhoons and southwesterly air currents traversing from the South China Sea to Taiwan are the main factors influencing debris flows. According to the CWB, between 1897 and 2018, an average of 6.8 typhoon warnings was issued each year in Taiwan [28]. One sudden meteorological change was observed in 2018: relatively fewer typhoons (only two) occurred and less rainfall was distributed in mountainous areas (Figure 5). The combined effect of typhoons and southwest current–induced rainfall caused a relatively large number of landslides. Four and five typhoons affected Taiwan but did not make landfall in 2009 and 2020, respectively.

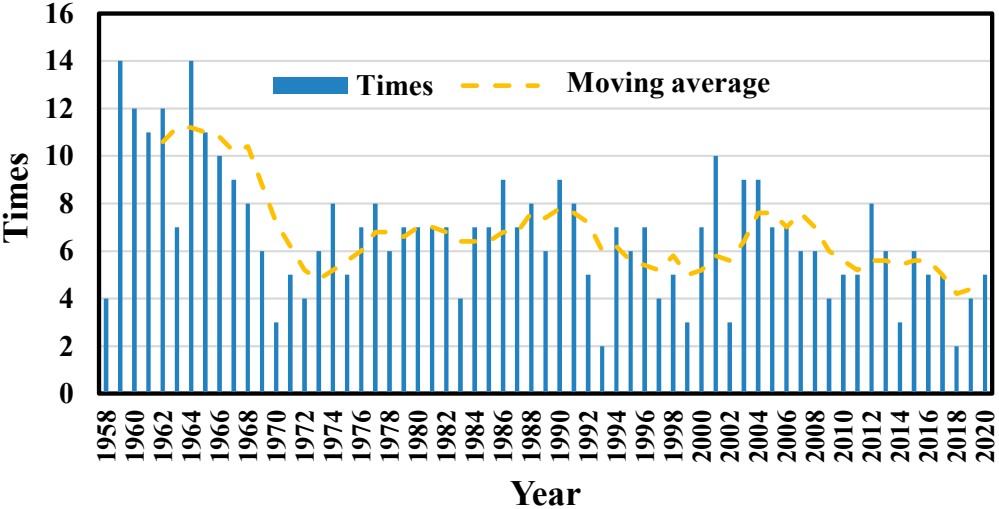

**Figure 5.** Number of typhoons with warnings issued in Taiwan (moving average: every 5 years).

The annual maximum rainfall intensity attributable to the typhoons exceeded 50 mm/h (Figure 6). The maximum intensity of 186 mm is ascribable to Typhoon Nakri in 2002 at Pengjia Islet (located north of Taiwan's main island) and 167.5 mm in 2012 by Typhoon Tembin in southern Taiwan. The data demonstrate 12 instances of maximum rainfall intensity of over 100 mm/h between 1980 and 2020. High rainfall intensity (>50 mm/h) is observable each year.

As shown in Figure 7a, the average annual temperature was the highest (24.4 °C) and lowest (23.3 °C) in 2016 and 2011 in plain areas, respectively. Years with low temperatures corresponded to higher cumulative rainfall (e.g., in 2005, 2008, 2011, and 2012); however, this trend was not observed in 2002 or from 2014 through 2018. In 2016, an unusual change in the temperature and cumulative rainfall was noted: both attained their highest values in the same year. High temperatures and lower cumulative rainfall have been recorded since 2019.

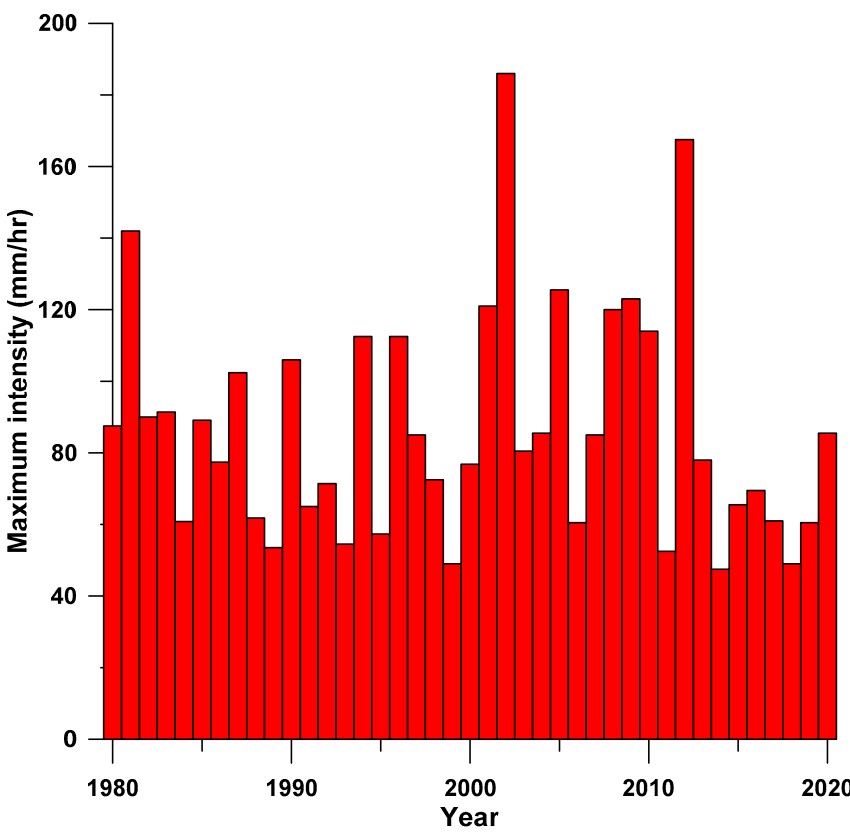

**Figure 6.** Maximum rainfall intensity caused by typhoons between 1980 and 2020 (Source: CWB Typhoon Database, https://rdc28.cwb.gov.tw/TDB/ (accessed date 30 May 2020)).

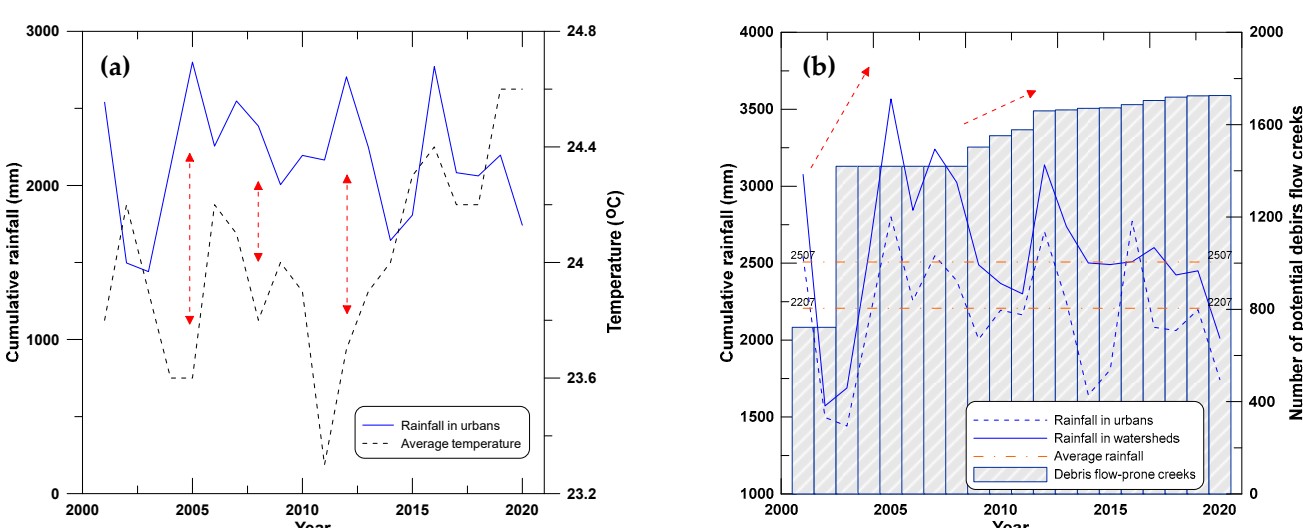

**Figure 7.** Statistics of average annual cumulative rainfall with respect to (**a**) temperature in urban areas and (**b**) debris flow–prone creeks in mountain watersheds between 2001 and 2020.

The average annual precipitation between 2011 and 2020 in urban areas was 2207 mm and only 1742.4 mm in 2020. In mountain watersheds, the average annual precipitation was 2507 mm but only 2012 mm in 2020. Rainfall in the mountain watershed areas was higher than that in urban areas. As presented in Figure 7b, in 2016, the average cumulative rainfall in urban areas was higher than that in the mountain watersheds. This phenomenon could be due to the relatively fewer occurrences of typhoons in 2016–2020 in that less rainfall was brought to the mountain watershed areas (Figure 5). The number of debris flow–prone

creeks increased sharply from 766 to 1420, coinciding with the increase in cumulative rainfall: from 1572 mm in 2002 to 2572 mm in 2005 and from 2300 mm in 2011 to 3139 mm in 2012.

To compare the distribution of rainfall induced by climate change in plains and mountain watershed areas, two rain gauge stations were selected for examination. The Xikou rain gauge station (elevation: 17 m) is located at the coastal area of the plain region in Chiayi County in southern Taiwan. The station recorded an average annual precipitation of 1379.3 mm between 1967 and 2020 [27]. Over the same period, Xiaogongtian station (elevation: 680 m) is situated in the same county received an average annual precipitation of 3180.5 mm. As presented in Figure 8a, the number of rainy days and annual cumulative rainfall were higher in the mountain watershed areas than in the urban areas. The average rainfall per rainy day (i.e., the cumulative rainfall divided by the number of rainy days) in the mountain watershed areas was higher between 2004 and 2009 and decreased considerably in the period from 2010 to 2020.

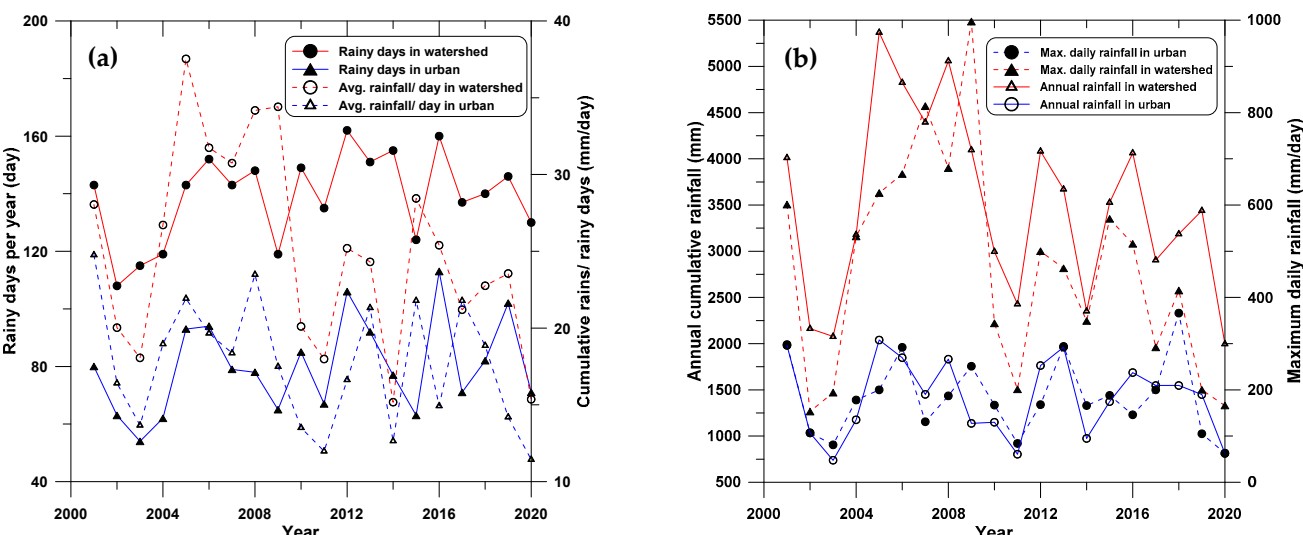

**Figure 8.** Annual changes between 2000 and 2020 in urban areas (Xikou station) and mountain watershed areas (Xiaogongtian station) with regard to the (**a**) number of rainy days and average rainfall per rainy days, and (**b**) cumulative rainfall and maximum daily rainfall.

As shown in Figure 8b, daily rainfall (up to 1000 mm) in 2009 peaked during Typhoon Morakot. However, the maximum daily rainfall did not coincide with the maximum yearly cumulative rainfall. This reflects the occurrence of torrential rain, which occurred on 25 August 2018 in urban areas in southern Taiwan, caused large-scale flooding. Climate change has resulted in higher temperatures, relatively fewer typhoons occurred and less rainfall was distributed in mountainous areas in recent (2018–2020) years.

### 3.2. Climate Change and Landslides

In general, the annual cumulative rainfall in mountain watersheds was proportional to the number of southwesterly flow-induced landslides (Figure 9). However, the number of typhoon-induced landslides was not proportional to the amount of yearly cumulative rainfall. Typhoon-induced rainfall was high in both intensity and cumulative rainfall. For example, in 2009, Typhoon Morakot was responsible for an annual cumulative rainfall of 2489 mm in the mountain watersheds, a number that is close to the average annual precipitation between 1949 and 2008 of 2502 mm [29]. Overall, typhoon-induced rainfall caused more landslides than did the meiyu front, tropical low pressure, or southwesterly flow events. The opposite observation applies to the 2017–2020 period, wherein the number of landslides decreased with fewer typhoons making landfall. This is because fewer

typhoons making landfall led to a reduction in typhoon-induced rainfall, especially in mountain watershed areas in mountainous regions.

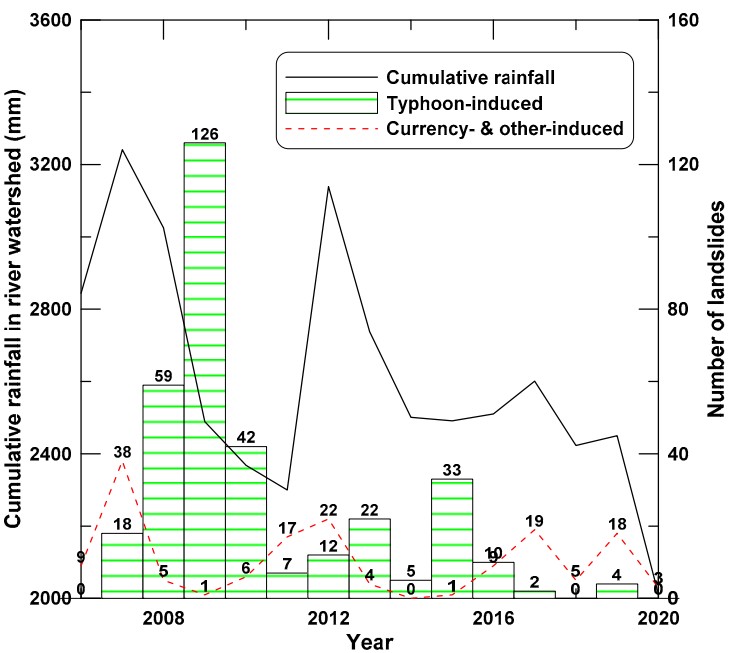

**Figure 9.** Number of typhoons and other extreme weather event-induced landslide disasters and annual cumulative rainfall in basin areas.

### 3.3. Rainfall Thresholds for Landslide Initiation

Based on the regression characteristics by Im-Ac plot (Figure 10), B-1 and C-1 regression lines separate the data into three groups. A three rainfall characteristics that trigger landslides are high rainfall intensity over a short duration (<12 h; group A), high intensity and high cumulative rainfall (group B), and high cumulative rainfall over a long duration (>36 h; group C), as displayed in Figure 10. Although Im and Ac are mutually dependent and related to D, all regression lines show a favorable $r^2$ value of over 95%. The exception is the regression line A-1, which exhibits a considerable separation of rainfall duration over a 12-h period based for rainfall characteristics of landslide initiation.

The landslide data were plotted in the Ac–D and Im–D modes (Figure 11). Clearly, Taiwan's slopeland areas frequently receive rainfall that is prolonged (up to 5 days) and high in cumulative rainfall (>1000 mm). From 2018 to 2020, landslides were initiated at a lower Ac (<300 mm) and Im (<10 mm/h; Figure 3). Instantaneous high-intensity rainfall and high daily cumulative rainfall are common because of climate change. The lower bound of the regression line in Figure 11a for an Ac of <100 mm in the Im–D plot represents the characteristics of instantaneous high-intensity rainfall that can trigger landslides:

$$Im = 36.69D^{-0.8} \quad (r^2 = 0.81 \text{ for an Ac of } <100 \text{ mm}) \tag{3}$$

Regarding landslides triggered by a high daily cumulative rainfall (over 24 h), the regression line for the Ac–D plot is presented in Figure 11b. The corresponding equation is as follows:

$$Ac = 37.54D \quad (r^2 = 0.99, \text{ for Im} > 30 \text{ mm/h}) \tag{4}$$

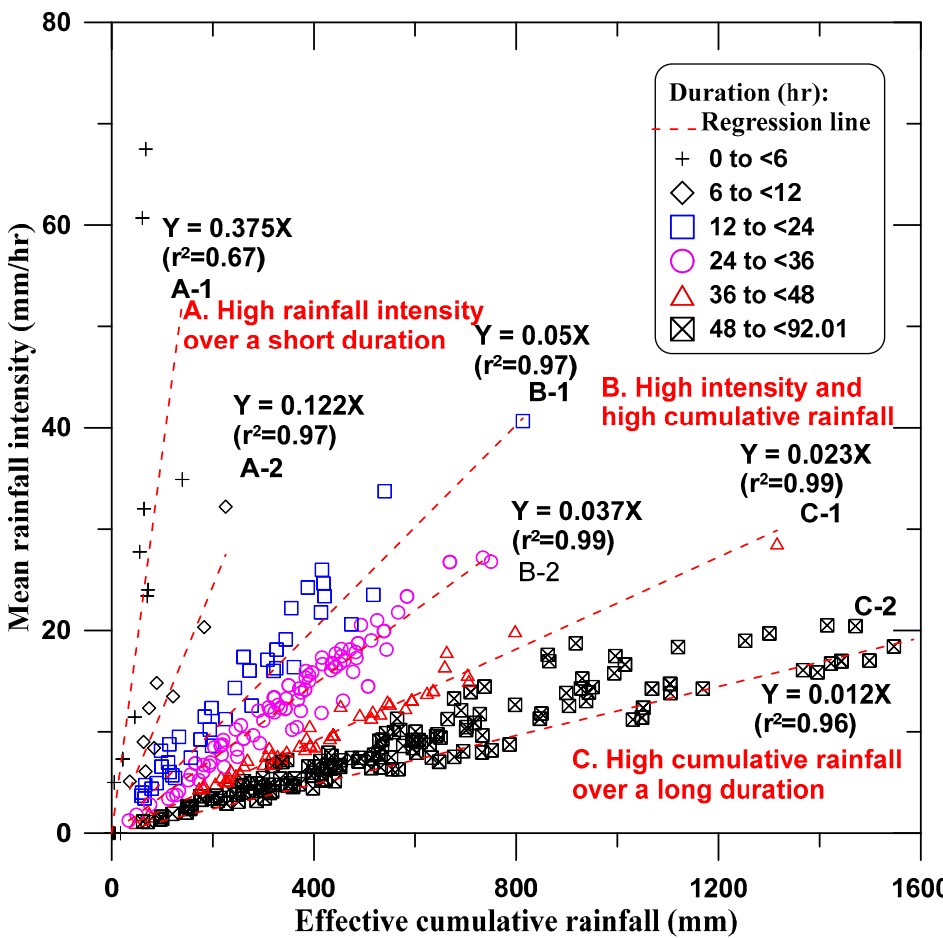

**Figure 10.** Three rainfall characteristics that trigger landslides: high rainfall intensity over a short duration (<12 h; group A), high intensity and high cumulative rainfall (group B), and high cumulative rainfall over a long duration (>36 h; group C).

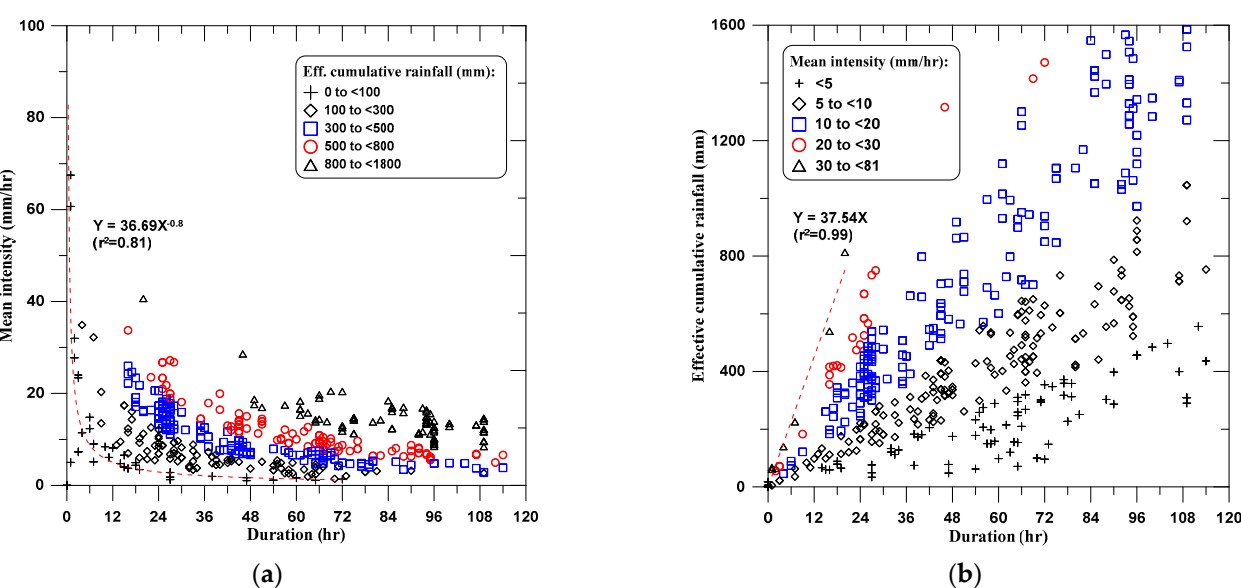

**Figure 11.** Rainfall characteristics that trigger landslides in the (**a**) Im–D plot and (**b**) Ac–D plot.

## 4. Discussion

### 4.1. Typhoon-Induced and Non-Typhoon-Induced Landslides

Climate change is causing less typhoon-brought rainfall in Taiwan and a distinction is needed between typhoon-induced and non-typhoon-induced landslides. In examining Figure 12a,b, it is clear that more landslides (341) were triggered by typhoon-induced rainfall than by non-typhoon-induced rainfall (145), respectively. The number of landslides triggered by typhoon-induced rainfall that was prolonged and high in cumulative rainfall was greater than that triggered by non-typhoon-induced rainfall. This is attributable to the fact that typhoons cause rainfall of higher intensity and higher cumulative rainfall over wider areas than do non-typhoon events. Landslides triggered by typhoon-induced rainfall occurred at an Im of <20 mm/h and a D of >12 h (Figure 12c), whereas those triggered by non-typhoon-induced rainfall occurred at an Im of <10 mm/h (Figure 12d).

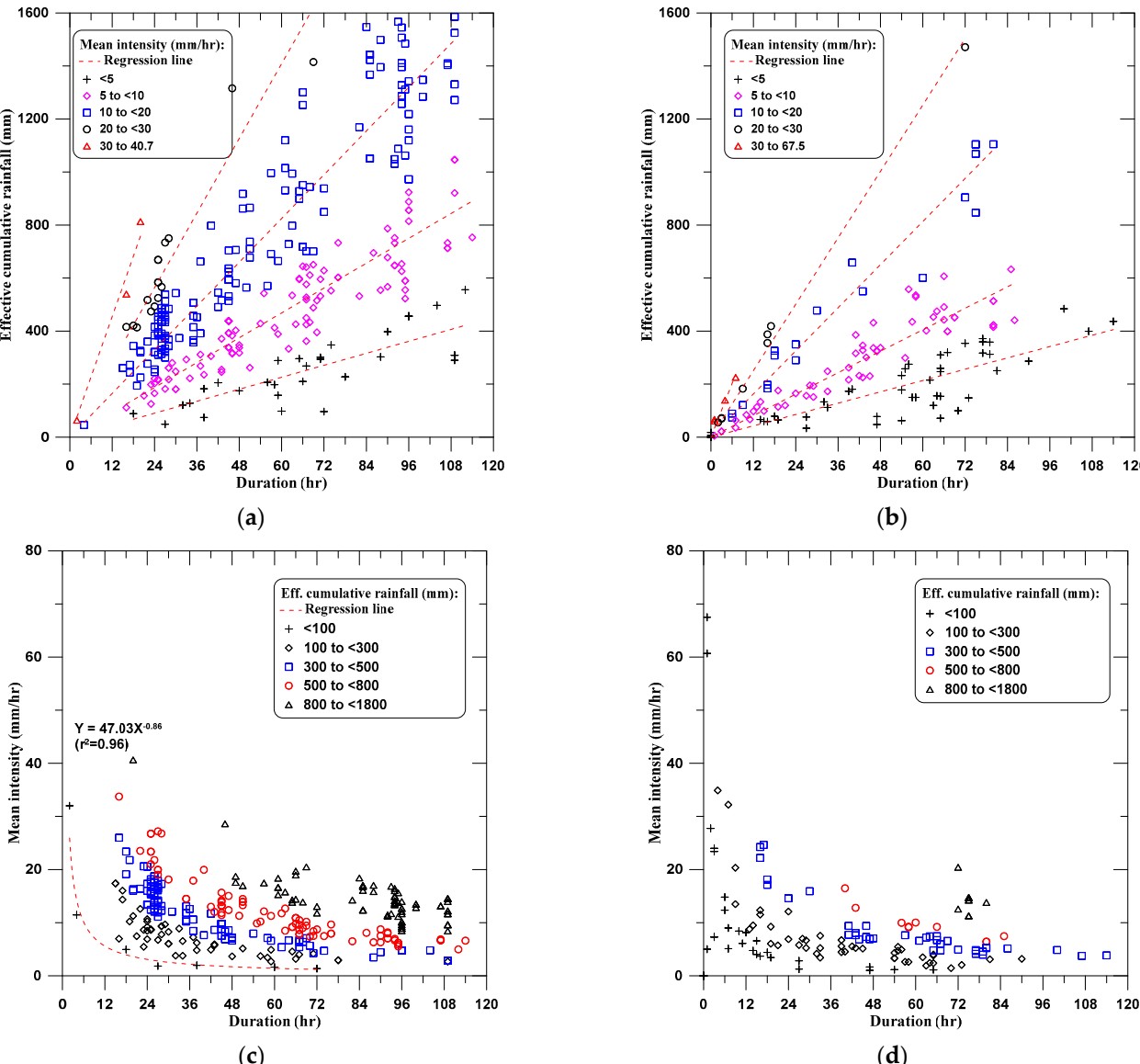

**Figure 12.** Rainfall characteristics of landslide initiation for (**a**) typhoon-induced landslides in the Ac–D plot, (**b**) non-typhoon-induced landslides in the Ac–D plot, (**c**) typhoon-induced landslides in the Im–D plot, and (**d**) non-typhoon-induced landslides in the Im–D plot.

The results demonstrate that non-typhoon-induced landslides were initiated within a relatively short duration (12 h) and under a cumulative rainfall of $\leq$200 mm. Typhoon-induced landslides had a higher rainfall threshold and occurred over a longer duration (>18 h).

### 4.2. Differences between Landslides and Channelized Debris Flows

The 152 channelized debris flows were separated from the 334 falls, slides, shallow/deep landslides, and other types of landslides for analysis (Figure 13). No major differences were observed among the landslides and channelized debris flows triggered at an Im of 5–20 mm/h, but a notable difference was detected for such events triggered at an Im of >20 mm/h (Figure 13a vs. Figure 13b). The upper bound for landslide initiation corresponding to a D of <18 h and an Im of >30 mm/h (Figure 13b) is:

$$Ac = 39.677D \ (r^2 = 0.99) \tag{5}$$

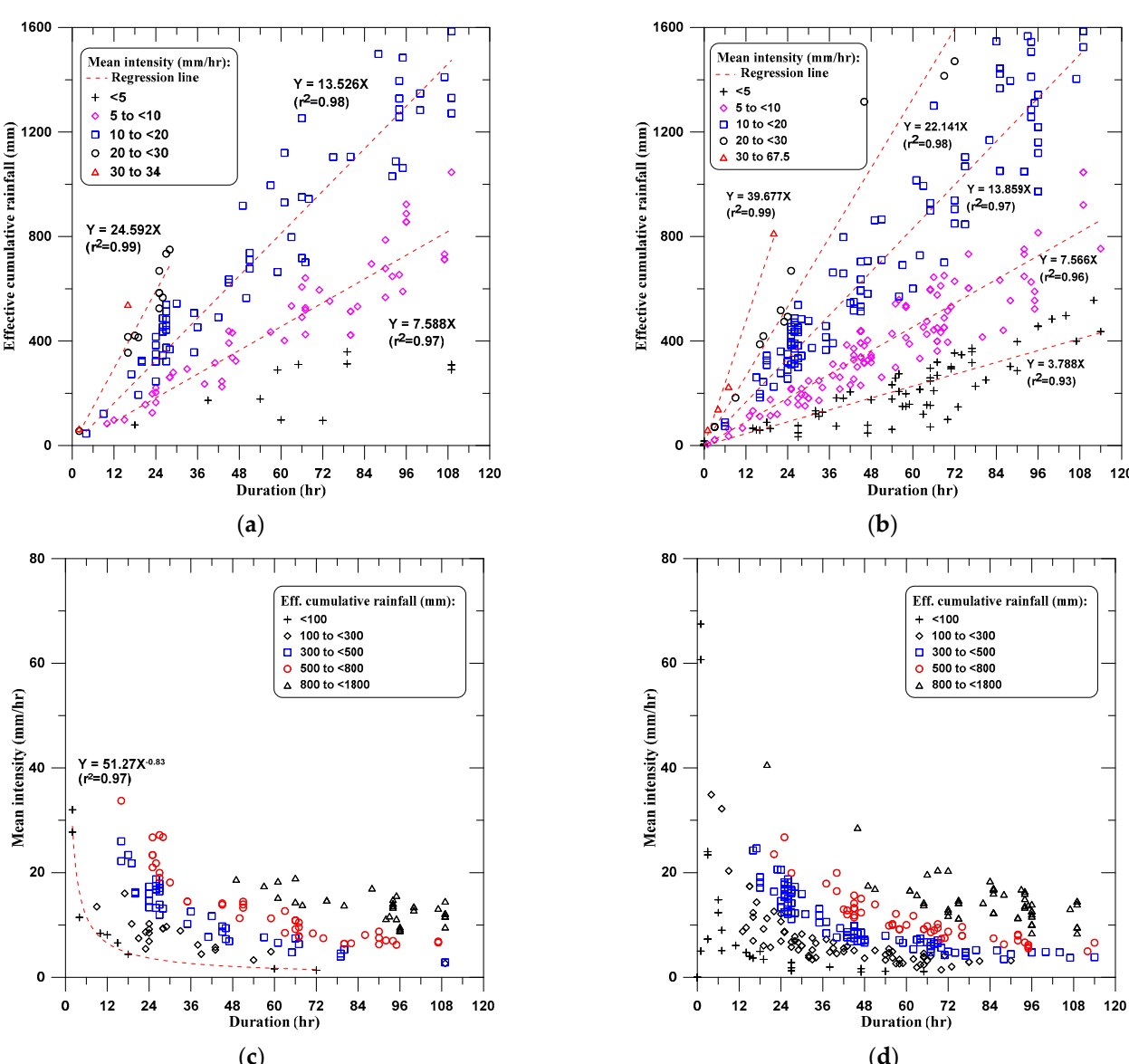

**Figure 13.** Rainfall characteristics in the model of the (**a**) Ac–D plot to initiate channelized debris flows; (**b**) Ac–D plot to initiate falls, slides, shallow/deep landslides, or other types of landslides; (**c**) Im–D plot to initiate channelized debris flows; and (**d**) Im–D plot to initiate falls, slides, shallow/deep landslides, and other types of landslides.

Low rainfall intensity corresponding to an Im of <10 mm/h and a D of <12 h was less likely to trigger channelized debris flows (Figure 13c). Low rainfall intensity (Im < 5 mm/h) took a longer duration to initiate channelized debris flows. By contrast, landslides were initiated over a short duration (Figure 13d). The lower threshold bound for initiating channelized debris flows under an Ac of <100 mm can be presented as:

$$\text{Im} = 51.27 D^{-0.83} \ (r^2 = 0.97) \tag{6}$$

### 4.3. Warning Model and Applications

Considering the aforementioned results, landslide warning models were suggested in an Im–D plot with Ac classification (Figure 14) and in an Ac–D plot with Im classification (Figure 15). The Im–D model can be used for the real-time monitoring of rainfall intensity. The Ac–D model can be employed in determining the cumulative rainfall threshold of slopeland areas at which landslide warnings are issued. For short-duration, high-intensity rainfall caused by climate change, the upper bound at which a landslide trigger warning is issued can be expressed as:

$$\text{Ac} = 28.784 D \ (r^2 = 0.98) \tag{7}$$

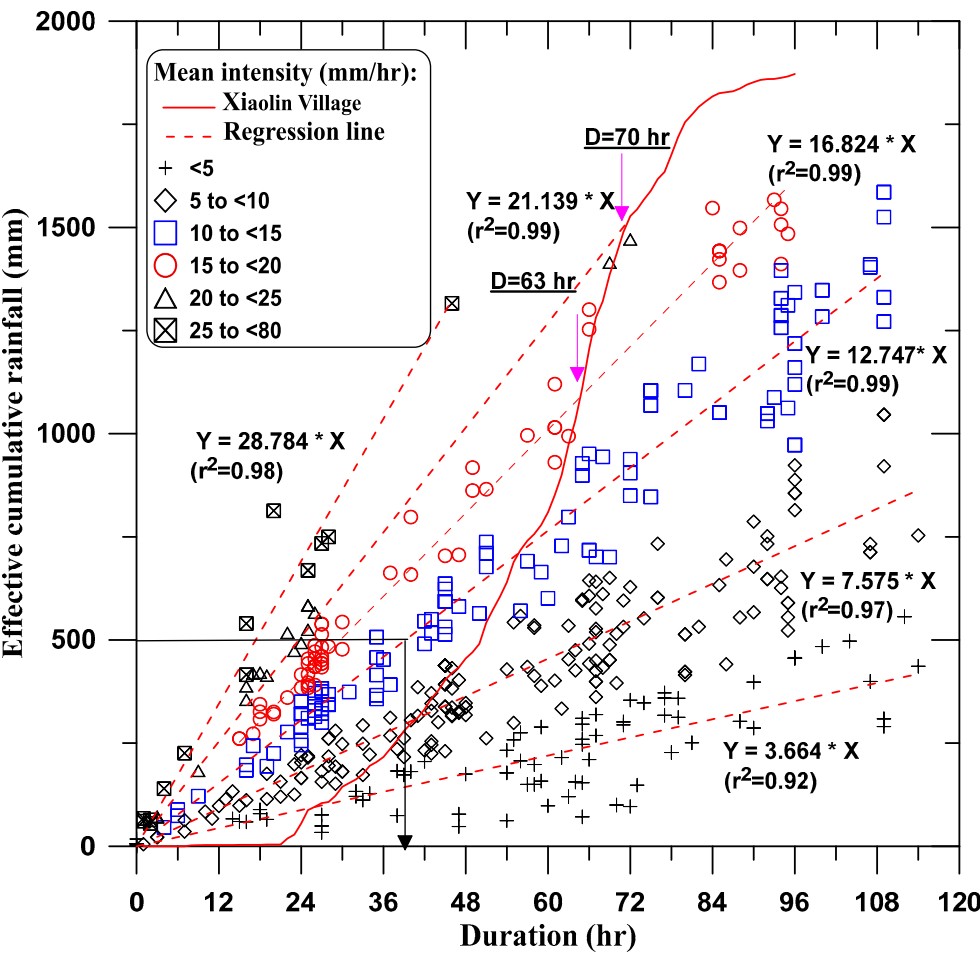

**Figure 14.** Relationship of effective cumulative rainfall and duration with changes in the mean rainfall intensity threshold at which landslides are initiated.

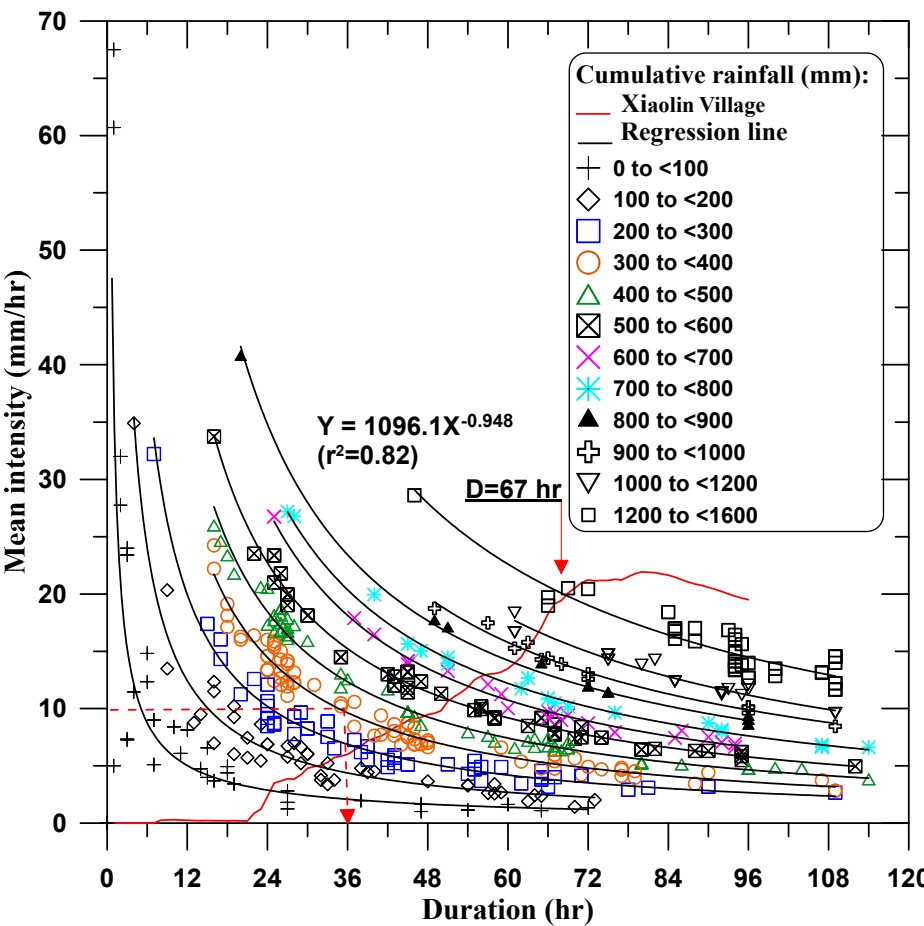

**Figure 15.** Relationship of rainfall intensity and duration with changes in the effective cumulative rainfall threshold at which landslides are triggered.

Herein, two cases were applied to the proposed warning model for a complex disaster: (1) channelized debris flow and breaching of a dam by a landslide during Typhoon Morakot in 2009 and (2) channelized debris flow during a frontal rainfall event in 2019.

The landslide dam breach led to flash floods and debris flow in Xiaolin Village in southern Taiwan, causing 465 residents to be buried by debris masses during Typhoon Morakot in 2009 [30]. The channelized debris flow occurred from 1:00 AM on 6 August 2009, to 19:00 AM on 8 August 2009 after 67 h of rainfall, where the Im was 19.1 mm/h and the cumulative rainfall was 1281.5 mm. Following the debris flow, a large-magnitude landslide was triggered at 06:00 AM on 9 August 2009 after 78 h of rainfall, where the Im was 21.5 mm/h and the cumulative rainfall was 1676.5 mm [31].

A combination model was employed to estimate the time of channelized debris flow and the subsequent landslide. Figure 14 presents the Ac–D plot of rainfall following the regression line of Im = 15–20 mm/h. Precipitation increased to up to 1000 mm in the 63rd hour. For an Ac of >1500 mm and an Im of >20 mm/h, the time taken to reach the upper bound of the regression line was 70 h. In Figure 14, the input of Im = 20 mm/h and Ac > 1200 mm correspond to a D of 67 h.

Torrential rainfall induced channelized debris flow on 15 August 2019 in Lougui Village in Kaohsiung County in southern Taiwan. The rainfall began at 00:00 on August 15 and triggered debris flow at 10:00 on August 16 over a 34-h duration. The debris flow occurred at an Ac of 515 mm under an Im exceeding 10 mm/h [32].

The input of Ac = 500 mm and Im = 10–15 mm/h correspond to a D of 39 h (Figure 14); alternatively, D = 36 h for Ac = 500 mm and Im = 10–20 mm/h (Figure 13a). According to the SWCB, the rainfall threshold at which landslide warnings can be issued is 350 mm; the

input of Im = 10 mm/h correspond to a D of 36 h (Figure 15). These estimates of rainfall durations that can initiate debris flows are close to time of triggered debris flow and are therefore valuable for the issuance of early warnings.

The model can be integrated with finite element analysis of rainfall seepage on slopes to determine the cumulative rainfall threshold at which landslides are triggered and to determine the corresponding rainfall duration threshold at which warnings are issued (Figure 14). Furthermore, real-time monitoring can be employed to determine the rainfall duration for a specified cumulative rainfall (Figure 15).

## 5. Conclusions

Global climate change has influenced the climate of Taiwan. The climate change trends in Taiwan demonstrate that temperatures in urban areas have increased by 1 °C since 1998. Rainfall distributions in plain areas are close to the average distribution, but precipitation in mountain watershed areas is lower than the average annual values. Typhoon-induced rainfall caused fewer landslides than did the meiyu front, tropical low pressure, or southwesterly flow events, and the number of landslides decreased with the reduction in typhoon events. Three rainfall characteristics that trigger landslides were identified: high-intensity rainfall over a short duration (<12 h), rainfall that is high in intensity and cumulative rainfall, and high cumulative rainfall over a long duration (>36 h). Climate change has led to higher temperatures, less precipitation in mountain watershed areas, and a lower rainfall threshold at which landslides are triggered by non-typhoon climate events. Herein, combinations of warning models for landslides in cumulative rainfall–duration plots with rainfall intensity classification and a mean rainfall intensity–duration plot with classification in cumulative rainfall were proposed.

**Author Contributions:** Conceptualization, H.-W.C. and C.-Y.C.; methodology, H.-W.C.; software, C.-Y.C.; validation, C.-Y.C.; formal analysis, H.-W.C.; investigation, C.-Y.C.; resources, H.-W.C.; data curation, H.-W.C.; writing—original draft preparation, C.-Y.C.; writing—review and editing, C.-Y.C.; visualization, H.-W.C. All authors have read and agreed to the published version of the manuscript.

**Funding:** This research received no external funding.

**Institutional Review Board Statement:** Not applicable.

**Informed Consent Statement:** Not applicable.

**Data Availability Statement:** The data of debris flow prone creeks and landslides for this article are available in a data repository in https://246.swcb.gov.tw/. Rainfall data are available in a data repository in https://gweb.wra.gov.tw/ and https://www.cwb.gov.tw/Data, accessed on 19 January 2022.

**Acknowledgments:** The author thanks the Soil and Water Conservation Bureau (SWCB), Water Resource Agency (WRA), and Central Weather Bureau (CWB) in Taiwan for providing valuable materials for this study and express gratitude to the reviewers for their useful comments.

**Conflicts of Interest:** The authors declare no conflict of interest.

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
