# Peer review of "Warning Models for Landslide and Channelized Debris Flow under Climate Change Conditions in Taiwan"

_water, doi:10.3390/w14050695_

Round 1
Reviewer 1 Report
In the manuscript titled “Local Climate Change and Landslide Initiation in Taiwan”, the authors show a case-study regarding the statistical analysis of the local climate change and landslides in Taiwan. So, the authors carried out a careful study on the climate trend (rainfall and temperature) in Taiwan. The achieved results highlighted that in central Taiwan the annual precipitation in watersheds exhibit a declining trend and the higher number of landslides were triggered mainly by typhoon-induced rainfall.
The main criticism of the manuscript is that the research appears like to be a too local study. So, in my opinion, the manuscript is not interesting for an international reader that doesn't live in Taiwan. At this regard, I would suggest to increase the international interest of the manuscript, for example a comparison between the achieved data and other similar researches in worldwide would be welcomed. Indeed, there are many researches on change climate in worldwide, so a correlation between these ones and the achieved results in this research is needed. Furthermore, the authors should improve the description regarding the landslide phenomena and rainfall induced landslides adding some photos on debris flows. In this version of the manuscript the authors show only a generic geomorphological information on landslides.
Other critical aspect is that the manuscript is lacking of a “Discussion data” section where the authors should make an indepth and critical discussion of the achieved results. In this new section, the authors should also emphasize what is the new knowledge and what is already known to the international scientific community in their research.
Finally, a revision of the English form is recommended.
For these reasons, I retain that the manuscript in this current form cannot be published in the Water Journal and there are some issues to be worked out. Therefore, in my opinion, the manuscript needs a Major Revision. I wish that my notes are useful to the authors to improve their manuscript and to bring it up to publication standards.
Author Response
Response to Reviewer 1:
In the manuscript titled “Local Climate Change and Landslide Initiation in Taiwan”, the authors show a case-study regarding the statistical analysis of the local climate change and landslides in Taiwan. So, the authors carried out a careful study on the climate trend (rainfall and temperature) in Taiwan. The achieved results highlighted that in central Taiwan the annual precipitation in watersheds exhibit a declining trend and the higher number of landslides were triggered mainly by typhoon-induced rainfall.
The main criticism of the manuscript is that the research appears like to be a too local study. So, in my opinion, the manuscript is not interesting for an international reader that doesn't live in Taiwan. At this regard, I would suggest to increase the international interest of the manuscript, for example a comparison between the achieved data and other similar researches in worldwide would be welcomed. Indeed, there are many researches on change climate in worldwide, so a correlation between these ones and the achieved results in this research is needed. Furthermore, the authors should improve the description regarding the landslide phenomena and rainfall induced landslides adding some photos on debris flows. In this version of the manuscript the authors show only a generic geomorphological information on landslides.
Response:
We delete the term “local” as suggested by the anymous reviewer. (Reviewer 3: The use of the word "local" is not necessary. Local, regional or global, climate change is happening at all scales.) The model is first proposed and could be used worldwide for a practical application.
The title of the manuscript is changed as “Warning Models for Landslide and Channelized Debris Flow under Climate Change in Taiwan.”
A new added Figure 2 has been added for cases of rainfall-induced landslide and channelized debris flow.
In P3, we explained that: The landslides were mainly classified as shallow and deep slides, falls, and channelized debris flows. The triggering mechanisms of landslides are different from channelized debris flows (creek). The available warning models are not well separated for issuing the different trigger mechanisms of landslides.
Other critical aspect is that the manuscript is lacking of a “Discussion data” section where the authors should make an indepth and critical discussion of the achieved results. In this new section, the authors should also emphasize what is the new knowledge and what is already known to the international scientific community in their research.
Response:
The triggering mechanisms of landslides are different from channelized debris flows. The available warning models are not well separated for issuing the different trigger mechanisms of landslides. The manuscript aims to improve the current literatures of warning models by considering climate change on the impacts of landslides.
Finally, a revision of the English form is recommended.
For these reasons, I retain that the manuscript in this current form cannot be published in the Water Journal and there are some issues to be worked out. Therefore, in my opinion, the manuscript needs a Major Revision. I wish that my notes are useful to the authors to improve their manuscript and to bring it up to publication standards.
Response:
The manuscript has been proofreading by native speakers (see attached English editing certificate). A major revision marked changes in red color texts in the manuscript.
Reviewer 2 Report
This paper studies the relationship between climate change (including the evolution of rainfall, typhoon, and air temperature) and landslides initiation in Taiwan. This issue is a very interesting topic, and the results of the research are of great significance to guide us in the prevention of natural induced disasters. However, there still exist some issues should be addressed before I can recommend this article. General and specific comments are given below.
GENERAL COMMENTS
1. There is a correlation between the frequency of landslides and climate changes (such as temperature rise and fall), whether the correlation is positive or negative, it exists objectively. However, it must be pointed out that correlation is not necessarily causation. Correlation is a necessary but not sufficient condition for causation. The authors allude to this causal relationship (especially the relationship between temperature and landslide initiation), making the paper somewhat flawed in the scientific method. It is suggested to make this article more rigorous by declaring differences in causation and correlation.
2. The Section of method is very important. However, current description in this section is too simplistic.
3. References. Regarding references, recently, there have been several latest papers closely related to your current study. The following references, but not limited, can be well analysed to strengthen your research:
[1]. Chen, P.-Y.; Tung, C.-P.; Tsao, J.-H.; Chen, C.-J. Assessing Future Rainfall Intensity–Duration–Frequency Characteristics across Taiwan Using the k-Nearest Neighbor Method. Water 2021, 13, 1521.
[2]. Chen, Y.-M.; Chen, C.-W.; Chao, Y.-C.; Tung, Y.-S.; Liou, J.-J.; Li, H.-C.; Cheng, C.-T. Future Landslide Characteristic Assessment Using Ensemble Climate Change Scenarios: A Case Study in Taiwan. Water 2020, 12, 564.
[3]. Chien, L.-K.; Hsu, C.-F.; Yin, L.-C. Warning Model for Shallow Landslides Induced by Extreme Rainfall. Water 2015, 7, 4362-4384, doi:10.3390/w7084362.
[4]. Du, Y.; Xu, Q.; Zhang, L.; Feng, G.; Li, Z.; Chen, R.-F.; Lin, C.-W. Recent Landslide Movement in Tsaoling, Taiwan Tracked by TerraSAR-X/TanDEM-X DEM Time Series. Remote Sensing 2017, 9, 353.
[5]. Lin, G.-W.; Kuo, H.-L.; Chen, C.-W.; Wei, L.-W.; Zhang, J.-M. Using a Tank Model to Determine Hydro-Meteorological Thresholds for Large-Scale Landslides in Taiwan. Water 2020, 12, 253.
[6]. Liu, J.-K.; Shih, P.T.Y. Topographic Correction of Wind-Driven Rainfall for Landslide Analysis in Central Taiwan with Validation from Aerial and Satellite Optical Images. Remote Sensing 2013, 5, 2571-2589.
SPECIFIC COMMENTS
4. The resolution of Figure 1 is too low, and the relevant legend characters are too small to be noticed. It is recommended to enchance the Figure 1.
5. Figure 6b shows the accumulated number of potential debris flow creeks. Can annual active debris flow creeks be shown each year? Current shown is that the number of debris flow creeks is increasing year by year. Considering that the active number should be different every year, what we prefer to see is the annual number of active debris flow creeks.
Author Response
Response to Reviewer 2:
This paper studies the relationship between climate change (including the evolution of rainfall, typhoon, and air temperature) and landslides initiation in Taiwan. This issue is a very interesting topic, and the results of the research are of great significance to guide us in the prevention of natural induced disasters. However, there still exist some issues should be addressed before I can recommend this article. General and specific comments are given below.
GENERAL COMMENTS
1. There is a correlation between the frequency of landslides and climate changes (such as temperature rise and fall), whether the correlation is positive or negative, it exists objectively. However, it must be pointed out that correlation is not necessarily causation. Correlation is a necessary but not sufficient condition for causation. The authors allude to this causal relationship (especially the relationship between temperature and landslide initiation), making the paper somewhat flawed in the scientific method. It is suggested to make this article more rigorous by declaring differences in causation and correlation.
Response:
- We add more information to strength the correlations of landslide and climate change.
In P1, The changing climate is rising global temperatures, shifting rainfall patterns, and more heavy rainstorms and record high temperatures [2].
Rising temperatures will increase evaporation that result in more frequent and intense storms [3]. Storm-affected areas are likely to experience increases in precipitation and increased risk of landslide.
[2] EPA, 2022. U.S. Environmental Protection Agency, https://www.epa.gov/climate-indicators (accessed on 8 February 2022).
[3] GPM, 2022. The Global Precipitation Measurement, https://gpm.nasa.gov/resources/faq/how-does-climate-change-affect-precipitation (accessed on 8 February 2022)
- The Section of method is very important. However, current description in this section is too simplistic.
Response:
In P4, the following paragraphs have been added to strengthen the method used.
Linear and nonlinear best-fit regression analyses were used to determine the regression lines for reflecting the rainfall characteristics that trigger landslides and channelized debris flows.
Warning models for landslides in cumulative rainfall–duration plots with rainfall intensity classification and mean rainfall intensity–duration plots with cumulative rainfall classification. The potential application of the proposed rainfall warning model was evaluated using the Xiaolin landslide case and a channelized debris flow case in southern Taiwan. The objective of this study was to identify the spatial distribution of the rainfall characteristics that trigger landslides and channelized debris flows.
- References. Regarding references, recently, there have been several latest papers closely related to your current study. The following references, but not limited, can be well analysed to strengthen your research:
[1]. Chen, P.-Y.; Tung, C.-P.; Tsao, J.-H.; Chen, C.-J. Assessing Future Rainfall Intensity–Duration–Frequency Characteristics across Taiwan Using the k-Nearest Neighbor Method. Water 2021, 13, 1521. https://doi.org/10.3390/w13111521
[2]. Chen, Y.-M.; Chen, C.-W.; Chao, Y.-C.; Tung, Y.-S.; Liou, J.-J.; Li, H.-C.; Cheng, C.-T. Future Landslide Characteristic Assessment Using Ensemble Climate Change Scenarios: A Case Study in Taiwan. Water 2020, 12, 564. https://doi.org/10.3390/w12020564
[3]. Chien, L.-K.; Hsu, C.-F.; Yin, L.-C. Warning Model for Shallow Landslides Induced by Extreme Rainfall. Water 2015, 7, 4362-4384. https://doi.org/10.3390/w7084362
[4]. Du, Y.; Xu, Q.; Zhang, L.; Feng, G.; Li, Z.; Chen, R.-F.; Lin, C.-W. Recent Landslide Movement in Tsaoling, Taiwan Tracked by TerraSAR-X/TanDEM-X DEM Time Series. Remote Sensing 2017, 9, 353.
[5]. Lin, G.-W.; Kuo, H.-L.; Chen, C.-W.; Wei, L.-W.; Zhang, J.-M. Using a Tank Model to Determine Hydro-Meteorological Thresholds for Large-Scale Landslides in Taiwan. Water 2020, 12, 253.
[6]. Liu, J.-K.; Shih, P.T.Y. Topographic Correction of Wind-Driven Rainfall for Landslide Analysis in Central Taiwan with Validation from Aerial and Satellite Optical Images. Remote Sensing 2013, 5, 2571-2589.
Response:
The first two papers related to the contents of the study have been cited in the manuscript in P1.
The short-duration extreme rainfall events will become stronger, especially for 1-hr duration events in Taiwan (Chen et al. 2021).
The landslide magnitudes triggered by high-level typhoons (top 5%–15%) and extreme-level typhoons (top 5%) will increase by 125%–200% and 77%, respectively, under climate change in the Xindian River catchment in Taiwan (Chen et al. 2020).
SPECIFIC COMMENTS
4. The resolution of Figure 1 is too low, and the relevant legend characters are too small to be noticed. It is recommended to enhance the Figure 1.
Response:
We amplify the figure and change colors of the legend for better clear the image. In addition, all figures’ line thickness are redrew and text size are increased for clear presentation.
- Figure 6b shows the accumulated number of potential debris flow creeks. Can annual active debris flow creeks be shown each year? Current shown is that the number of debris flow creeks is increasing year by year. Considering that the active number should be different every year, what we prefer to see is the annual number of active debris flow creeks.
Response:
The purpose of the study is to propose an innovation rainfall warning model for landslides and channelized debris flows. There is no information on annual active debris flow creeks available. However, the increments of annual potential debris flow creeks can show the increments of active debris flow (Fig. 6)
Reviewer 3 Report
The study analyses the impacts of climate change on the increased occurrences of landslides in the Taiwan region of China.
I would like to thank the authors for conducting a very good and rigorous study. However, there are some concerns that would like the authors to respond to and include in the revised manuscript.
Abstract:
After discussing the world, include a sentence about china. Thereafter move towards Taiwan.
The use of the word "local" is not necessary. Local, regional or global, climate change is happening at all scales. If you are discussing the climate change of Taiwan, simply putting "climate change" is enough even if you have used local datasets to analyze the phenomenon.
I would suggest the change in the title to "Assessing the relationship of changing climate with increased frequency of landslide occurrences in Taiwan, China".
The last sentence of the Abstract should be the conclusive results from this work.
Introduction:
Ln 31-36 Check these lines, they don't follow the context.
Ln 37 What changes have brought the "changes in the precipitation". Please clarify and reframe the sentence.
Ln 38 Sentence not clear
The last sentence of the introduction must highlight the aims and objectives of the present study in light of the problems in Taiwan and why this study is important.
Materials and methods
Ln 95-96 These sentences are not standard ways to describe the use of GIS
Ln 95 There are various interpolation techniques, which one has been used in this study, please clarify why only that has been used.
Results
3.1 Change of sub-heading to "indicators of climate change in Taiwan"
Ln 123, don't urban areas come in watersheds? Clarify.
Ln 139-142 Sources/ references?
Ln 145-146 Sources/ references?
Ln 179 Claim needs to be substantiated with evidence.
The whole concept of climate change in Taiwan as presented by the authors is very vague. As per the standard definition of climate change, the variation in weather parameters studied over more than 30 years is conventionally referred to as climate change. However, the study only analyses the temperature and precipitation data from 2000-2020, approximately 20 years.
This is climate variability and not climate change. The need to redo the manuscript or substantiate their claims with sources and references.
While discussing, "climate change in Taiwan" as the authors refer, a lot of disentangled facts and figures are provided, with no coherence. The authors are suggested to introduce the facts and figures coherently and summarise the main findings in the last paragraph of this subsection.
3.2 Climate change and landslides
Typhoons are natural weather phenomenon and relating them with landslide frequency, does it prove that climate change is causing increased landslides? Clarify
This section also needs a revamp
3.3. Rainfall Thresholds for Landslide Initiation
Ln 221 - 227, It seems the categories have been self-made, kindly either provide the source or discuss here a bit more about how you have arrived at these categories.
4. Discussion
Somewhere it needs to be emphasized why a distinction is needed for Typhoon-Induced and Non-Typhoon-Induced Landslides, please do it.
A lot of facts and figures without moving anywhere. A reader has this impression.
I suggest since the work is more oriented towards developing a rainfall-induced landslide early warning model, therefore the whole manuscript needs to be restructured focussing on this alone. Otherwise, the reader is lost in the facts and figures without any substantial conclusions.
I am suggesting a major revision and would be requesting the authors to kindly revise the manuscript according to the suggested revisions. I am looking forward to receiving the revised version as soon as possible.
Thanks
Author Response
Response to Reviewer 3:
The study analyses the impacts of climate change on the increased occurrences of landslides in the Taiwan region of China.
I would like to thank the authors for conducting a very good and rigorous study. However, there are some concerns that would like the authors to respond to and include in the revised manuscript.
Abstract:
After discussing the world, include a sentence about china. Thereafter move towards Taiwan.
The use of the word "local" is not necessary. Local, regional or global, climate change is happening at all scales. If you are discussing the climate change of Taiwan, simply putting "climate change" is enough even if you have used local datasets to analyze the phenomenon.
I would suggest the change in the title to "Assessing the relationship of changing climate with increased frequency of landslide occurrences in Taiwan, China".
The last sentence of the Abstract should be the conclusive results from this work.
Response:
- The manuscript title has been changed into “Warning Models for Landslide and Channelized Debris Flow under Climate Change in Taiwan” to better coincide with the special issue of the journal. The landslide frequency do not increase with climate change by the results of analysis.
- The following sentence has been moved to the end of Abstract for the conclusive result of the study.
Climate change has resulted in higher temperatures, less rainfall in mountain watersheds, and a lower rainfall threshold at which landslides are initiated under nontyphoon climate events.
Introduction:
- Ln 31-36 Check these lines, they don't follow the context.
Response:
The original L31-36 have been rewritten:
The changing climate is rising global temperatures, shifting rainfall patterns, and more heavy rainstorms and record high temperatures [2]. Rising temperatures will increase evaporation that result in more frequent and intense storms [3]. Storm-affected areas are likely to experience increases in precipitation and increased risk of landslide.
[2] EPA, 2022. U.S. Environmental Protection Agency, https://www.epa.gov/climate-indicators (accessed on 8 February 2022).
[3] GPM, 2022. The Global Precipitation Measurement, https://gpm.nasa.gov/resources/faq/how-does-climate-change-affect-precipitation (accessed on 8 February 2022)
- Ln 37 What changes have brought the "changes in the precipitation". Please clarify and reframe the sentence. Ln 38 Sentence not clear
Response:
The sentence has been revised as: Climate change increases the risk of extreme weather events, such as droughts, flooding, and heat waves [1].
- The last sentence of the introduction must highlight the aims and objectives of the present study in light of the problems in Taiwan and why this study is important.
Response:
The following sentence has been added at the end of Introduction section.
The objectives of the study are setting rainfall warning models to enhance the rainfall characteristics to trigger landslides and channelized debris flows under climate change by events collection, rainfall parameters measurement and statistical regression analysis.
Materials and methods
- Ln 95-96 These sentences are not standard ways to describe the use of GIS.
Ln 95 There are various interpolation techniques, which one has been used in this study, please clarify why only that has been used.
Response:
- The section Materials and methods has been implemented with more description of method used (P4)
- Spatial geographic information system analysis was performed using GIS package [24].
- The sentence of GIS interpolation techniques has been deleted.
Results
- 1 Change of sub-heading to "indicators of climate change in Taiwan"
Response:
Yes, the subheading has been revised as “indicators of climate change in Taiwan.”
- Ln 123, don't urban areas come in watersheds? Clarify.
Response:
In the manuscript, the term “watershed” has been replaced by “mountain watershed” to distinguish with “urban areas”.
- Ln 139-142 Sources/ references?
Response:
The sources of the data has been added to the sentence.
Notably, Taiwan’s topography strongly influences rainfall distribution (Fig. 3). From 1897–2020, the average annual precipitation in mountain watersheds was 2507 mm [27].
- Ln 145-146 Sources/ references?
Response:
The sources of the data has been added to the sentence:
Typhoons are the main source of rainfall in Taiwan, followed by the meiyu front, tropical low pressure, and southwesterly flow events [28].
- Ln 179 Claim needs to be substantiated with evidence.
Response:
At the end of the sentence, we add the evidence for the claim:
This phenomenon could be due to the relatively fewer occurrences of typhoons in 2016–2020 in that less rainfall was brought to the mountain watershed areas (Fig. 4).
- The whole concept of climate change in Taiwan as presented by the authors is very vague. As per the standard definition of climate change, the variation in weather parameters studied over more than 30 years is conventionally referred to as climate change. However, the study only analyses the temperature and precipitation data from 2000-2020, approximately 20 years. This is climate variability and not climate change. The need to redo the manuscript or substantiate their claims with sources and references.
Response:
- The climate data for the analysis are more than 30 yrs. For instance, in P5:
From 1900 to 2012, the average temperature in Taiwan increased by 1.25°C–1.5°C in 13 urban areas in plains [26].
From 1897–2020, the average annual precipitation in mountain watersheds was 2507 mm.
According to the CWB, between 1897 and 2018, an average of 6.8 typhoon warnings was issued each year in Taiwan [28].
Also, in P6, The data demonstrate 12 instances of maximum rainfall intensity of over 100 mm/h between 1980 and 2020.
- The data in Fig. 4 and Fig. 5 are sourced from 1958 and 1980, respectively. In the study, we are focus on the recent years with more considerable of climate change with landslides.
- While discussing, "climate change in Taiwan" as the authors refer, a lot of disentangled facts and figures are provided, with no coherence. The authors are suggested to introduce the facts and figures coherently and summarise the main findings in the last paragraph of this subsection.
Response:
The references are added in the manuscript to support the findings (see red color texts in the manuscript). At the end of this paragraph, we summarized the main findings of the section below:
Climate change has resulted in higher temperatures, relatively fewer typhoons occurred and less rainfall was distributed in mountainous areas in recent (2018-2020) years.
3.2 Climate change and landslides
- Typhoons are natural weather phenomenon and relating them with landslide frequency, does it prove that climate change is causing increased landslides? Clarify. This section also needs a revamp.
Response:
Climate change in Taiwan do not increase landslides and the title of the manuscript do not use the term “increasing”.
Climate change is not causing increased landslide frequency in Taiwan and that is not conclusive in the manuscript.
3.3. Rainfall Thresholds for Landslide Initiation
Ln 221 - 227, It seems the categories have been self-made, kindly either provide the source or discuss here a bit more about how you have arrived at these categories.
Response:
We revise this sentence as below:
Based on the regression characteristics by Im-Ac plot (Figure 9), B-1 and C-1 regression lines separate the data into three groups.
- Discussion
- Somewhere it needs to be emphasized why a distinction is needed for Typhoon-Induced and Non-Typhoon-Induced Landslides, please do it.
Response:
Climate change is causing less typhoon brought rainfall in Taiwan and a distinction is needed for typhoon-induced and nontyphoon-induced landslides, as explained in P10
- A lot of facts and figures without moving anywhere. A reader has this impression. I suggest since the work is more oriented towards developing a rainfall-induced landslide early warning model, therefore the whole manuscript needs to be restructured focusing on this alone. Otherwise, the reader is lost in the facts and figures without any substantial conclusions.
Response:
The early warning model is affected by the climate change in Taiwan and the manuscript is focusing on the effects of climate change on landslide. Both information on climate change and caused rainfall characteristics to initiate landslide are important.
I am suggesting a major revision and would be requesting the authors to kindly revise the manuscript according to the suggested revisions. I am looking forward to receiving the revised version as soon as possible.
Thanks
Response:
All changes of the context in the manuscript (excluding those delete parts and redrew figures) are marked in red color for better presenting these changes.
Round 2
Reviewer 1 Report
I read the new version of the manuscript titled: “Local Climate Change and Landslide Initiation in Taiwan”, authors: Ho-Wen Chen, Chien-Yuan Chen. The present version of the manuscript was sufficiently improved, so I retain that the manuscript, in the present form, can be considered for publication in Water Journal.
Reviewer 2 Report
In this round the authors responsed all five comments. All replies and revisions are acceptable. The author mainly made in-depth improvements in the abstract and Introduction chapter. The logic and readability of the paper are further enhanced. In view of the steady improvement of the paper, I recommend this article for Accepted.
Reviewer 3 Report
Thank you very much for including the suggestions provided on the earlier version of the manuscript.